

# New Particle Formation at a High Altitude Site in India: Impact of Fresh Emissions and Long Range Transport

**Vyoma Singla, Subrata Mukherjee\*, Adam Kristensson, Govindan Pandithurai, Kundan K. Dani, Vasudevan Anil Kumar**

Indian Institute of Tropical Meteorology, Pune

*\*Corresponding author: subrata.cat@tropmet.res.in*

**Abstract**

There is a lack of characterization of the aerosol population in Western India, how it is affected by meteorological parameters, and new particle formation and the influence on cloud condensation nuclei (CCN). For this reason, measurements of particle number size distribution, aerosol chemical composition, meteorology and cloud condensation nuclei number concentration were monitored at High Altitude Cloud Physics Laboratory (HACPL) in Mahabaleshwar mountain town in Western India between November 2016 and February 2017. Most air masses in this period originated from the Indian continent to the north-east of HACPL. New particle formation (NPF) events were observed on 47 days and mainly associated with these north-easterly air masses and high $SO_2$ emissions and biomass burning activities, while weaker or non-NPF days were associated with westerly air masses and relatively higher influence of local air pollution. The growth of newly formed particles enhanced the mass concentration of secondary organic and inorganic species of aerosol particles. The mean growth rate, formation rate, condensation sink and coagulation loss for the 13 strongest events was found to be $2.6 \pm 0.4$ nm $h^{-1}$, $2.8 \pm 1.4$ $cm^{-3}$ $s^{-1}$, $2.2 \pm 2.9$ $*10^{-2}$ $s^{-1}$ and $1.6 \pm 1.0$ $cm^{-3}$ $s^{-1}$ respectively. A closer examination of strong NPF events showed that low relative humidity and solar radiation favoured new particle formation. These NPF events lead to a significant increase in CCN concentration (mean ~$53 \pm 36$ %). The NanoMap method revealed that NPF took place up to several hundred kilometers upwind and to the north-east of HACPL.

## 1. Introduction

Atmospheric aerosols directly affect the global climate by altering the radiative balance of the Earth atmosphere system (Stier et al., 2007) and indirectly by altering the cloud properties (Fan et al., 2012). However, these effects depend on the particle number size distribution and chemical composition of aerosol particles. Recent studies have shown that particle size is more important than the chemical composition in cloud formation (Dusek et al., 2006; Rose et al., 2010). The particles in the upper Aitken mode (>50 nm diameter) and in the accumulation mode (100-1000 nm) have a substantial effect on the cloud properties and initiation of precipitation as cloud condensation nuclei (Rosenfeld et al., 2008). Further, the number concentration of accumulation mode particles is related to visibility degradation (See et al., 2006), and sub-micron particles with health related problems (Seaton and Denekamp, 2003).

New particle formation (NPF) is one of the major sources of sub-micron particles in the atmosphere. It is characterized by a significant enhancement in the number concentration of nucleation mode particles and subsequent growth of these nucleated particles due to condensation (Kulmala and Kerminen, 2008; Skrabalova et al., 2015). Typical formation rates (FR) of 3 nm particles are in the range 0.01-10 $cm^{-3}$ $s^{-1}$ and typical growth



rates (GR) in the range1-20 nm h$^{-1}$, depending upon temperature, humidity, pressure, and availability of
condensable vapours. In urban areas, FR is often higher ~100 cm$^{-3}$ s$^{-1}$ and even higher in the coastal areas and
industrial plumes ~ 10$^7$-10$^5$ cm$^{-3}$ s$^{-1}$ (Kulmala et al., 2004). The atmospheric observations and laboratory
experiments have identified sulfuric acid and organics as the two important precursors of NPF (Riipinen et al.,
2007; Barsanti et al., 2009). There is always a competition between the FR and GR of freshly nucleated particles
and the condensation sink and coagulation sink with larger particles (Vehkamki and Riipinen, 2012). As a result
of that, nucleation and Aitken mode particles smaller than 25 nm and between 25 and 100 nm diameter,
respectively tend to have a shorter lifetime than accumulation mode particles between 0.1 and 1 µm diameter. If
the FR is large enough and the condensation and coagulation sinks low enough during NPF, these recently
generated particles can survive long enough in the atmosphere to be transported over distances of several
hundred kilometers, depending on the wind speed (Kivekas et al., 2016).
The measurements of particle number size distribution show that NPF events and growth are widespread
(Kulmala et al., 2004; Kulmala and Kerminen, 2008; Gong et al., 2008; Yue et al., 2010). Many studies address
the spatial scale of NPF events in which the horizontal scale of NPF extends from hundreds to thousands of
kilometres (Kulmala et al., 2004; Stainer et al., 2004; Dal Maso et al., 2005; Yue et al., 2010, Kristenson et al.,
2014). The spatial scale of NPF events help to identify the origin of nucleation mode particles (Nemeth and
Salma, 2014; Väänänen et al., 2016; Shen et al., 2018). The newly formed particles become climatically
important when they attain the size of ~50 nm in diameter (Kerminen et al., 2012). The particles with diameter ≥
50 nm tend to act as CCN and affect the cloud microphysical and optical properties (Fiore et al., 2012). Model
studies show that NPF accounts for 5-50% of the CCN number concentration in the boundary layer (Spracklen
et al., 2008). A study by Wiedensohler et al.(2009) shows that the growing nucleation mode particles can
contribute to about 80% of the CCN number concentration during special conditions with strong NPF in China.
Several other studies report the significant enhancement in atmospheric CCN concentrations due to NPF events
(Levin et al., 2012; Creamean et al., 2011; Yue et al., 2011; Pierce et al., 2012; Ma et al., 2016; de Espana et al.,
2017). NPF is known to dominate the particle number concentrations in clean atmosphere as compared to
anthropogenically influenced regions (Lihavainen, 2003). However, the detailed mechanism of nucleation, its
growth and its connection with biogenic emissions, anthropogenic activities and atmospheric chemistry, and the
global role for CCN is still uncertain.
In this study, we identify the NPF events by analyzing particle number size distribution (PNSD) data at HACPL,
Mahabaleshwar between November 2016 and February 2017. The characteristics of PNSD, new particle
formation rate (FR), growth rate (GR) and condensation sink (CS) are evaluated during the identified NPF
events at HACPL. The role of meteorology and chemical composition of sub-micron non-refractory particulate
matter (NR-PM$_1$) during the growth of freshly nucleated particles is also analyzed. A cluster analysis is
performed to identify the geographical source areas responsible for the observed NPF events. The spatial scale
of NPF events and where the 1-2 nm diameter particles are formed around the station is also analyzed by using
the NanoMap method (Kristensson et al., 2014). Finally, the measured CCN number concentration, PNSD
(5.14-900 nm) and chemical composition of NR-PM$_1$ aerosol are used to evaluate the probable contribution of
NPF to the total CCN concentration at the sampling site.




## 2. Methodology

### 2.1 Measurement Site

Measurements were performed at HACPL (17.92° N, 73.65° E; 1378 m above mean sea level), located in the Western Ghats mountain range in south-west India. The site is at the small hill town Mahabaleshwar, which is also a tourist attraction. The town is surrounded by dense vegetation and residential houses, hotels and a rural market. During the study period, the site was influenced by both, local and regional anthropogenic pollution depending on the meteorological conditions and timing of source activities (Mukherjee et al., 2018).

### 2.2 Instrumentation

The aerosol chemical composition, particle number concentration and cloud condensation nuclei (CCN) number concentration were measured by using Time of Flight - Aerosol Chemical Speciation Monitor (ToF-ACSM), Wide Range Aerosol Spectrometer (WRAS) and CCN counter (CCNC) respectively. Due to different temporal resolutions of these instruments, the data sets were averaged hourly for the analysis. The QA/QC procedure for ToF-ACSM can be found in Mukherjee et al. (2018) and for WRAS and CCNC in Singla et al. (2017). The meteorological conditions (Solar radiation (SR), surface temperature (T), relative humidity (RH), wind speed (WS) and wind direction (WD)) were recorded using an Automatic Weather Station (AWS) at an interval of one minute.

#### 2.2.1 Time of Flight - Aerosol Chemical Speciation Monitor (ToF-ACSM)

A detailed description, specific to our ACSM instrument, its operation and calibration procedure is discussed in our earlier work (Mukherjee et al., 2018). In brief, before the ACSM, we sample the particle-laden and particle free air alternatively and then focus the particle beam into ACSM through an aerodynamic lens. In the detection chamber, the non-refractory particle fraction vaporizes at ~600∘C and ~$10^{-7}$ mbar and is subsequently ionized by the electron impaction ($E_{kin}$ = 70 eV) by a tungsten filament arranged perpendicular to the particle beam in the vaporization region. The ions are extracted by a set of ion optics and introduced into the TOF analyzer where they are orthogonally extracted and separated based on their mass-to-charge ratio. The lens system has almost 100 % transmission at vacuum aerodynamic diameters between 150–450 nm (Liu et al., 2007). The collection efficiency is assumed to be 0.5 in this analysis.

#### 2.2.2 Wide-range aerosol spectrometer (WRAS)

The aerosol particle number concentration (PNC) was measured as a function of particle size (range ~ 5 nm to 32 µm in 72 channels) at 4-minute interval by using WRAS (manufactured by Grimm, Germany). The detailed description and calibration procedure of WRAS can be found in Singla et al., 2017. The instrument is kept under air conditioning conditions for its running at constant temperature. The sampling probe features a Nafion dryer to reduce the humidity to ~40%, to avoid the effect of ambient humidity and to reduce the error in measuring diameter in the detection system. Although WRAS provides information in the size range of 5 nm - 32 µm, we have used PNC in the size range of 5.14 – 900 nm only to identify the NPF events in this study.

#### 2.2.3 Cloud Condensation Nuclei Counter (CCNC)



Our earlier work (Singla et al., 2017) gives the detailed description of the CCN counter used in this study. A
similar procedure was followed for the calibration of CCNC as discussed in Singla et al., 2017. The CCN
number concentration was monitored as a function of time and supersaturation (SS) using a single-column
continuous flow stream wise thermal gradient CCN chamber (DMT CCNC-100; Lance et al., 2006; Roberts and
Nenes, 2005). The instrument runs at a flow rate of 0.5 Lmin$^{-1}$ with a sheath-to-aerosol flow ratio of 10:1. The
activated particles are subsequently sized using an optical particle counter (OPC) and counted as CCN in the
diameter range of 0.75 – 10 µm. The instrument samples particles every second at five different SS (0.1, 0.3,
0.5, 0.7 and 0.9%). The CCN concentration at 0.1% SS is measured for 10 min and 5 min each for every other
SS. This gives one complete CCN SS spectra in half an hour. The CCN data was only used in the period when
the temperature was stable at each SS level.

**3. Data Analysis**
**3.1 Parameters characteristic of NPF events**
The criteria prescribed by Dal Maso et al.(2005) have been used for classifying the NPF events. The parameters
characteristic of NPF events - growth rate (GR), formation rate (FR), condensation sink (CS) and coagulation
loss (Coag) were calculated following the methodology of Dal Maso et al.(2005). The FR of 5nm particles was
calculated at the beginning of each event and is expressed as:

$$J_5 = \frac{dN_{5-25}}{dt} + F_{coag} + F_{growth}$$

where $N_{5-25}$ is the concentration of nucleation mode particles ($N_{nuc}$), $F_{coag}$ is the scavenging of particles due to
coagulation of freshly nucleated particles and $F_{growth}$ is the flux of particles growing out of the size range 5-25
nm. This growing flux of particles is defined as:

$$F_{growth} = \frac{1}{\Delta Dp} . GR_{5-25} . N_{5-25}$$

where $\Delta Dp$ is the change in diameter of particle i.e. 25nm - 5nm = 20 nm, in this case. $GR_{5-25}$ is the growth rate
of freshly nucleated particles in the size range ~5-25 nm. The GR was obtained by fitting a first-order
polynomial to the geometric mean diameters (GMD) of the nucleation mode particles. It is defined as:

$$GR_{5-25} = \frac{\Delta Dp}{\Delta t}$$

Here $\Delta D_p$ is the change in geometric mean diameter and $\Delta t$ is the time interval. Further, the coagulation loss
term, $F_{coag}$ is defined as:

$$F_{coag} = N_{5-25} . Coag_{5-25}$$
and $$Coag_{5-25} = \sum_j K_{ij} N_j$$



Coag$_{5\text{-}25}$ is the coagulation loss rate. K$_{ij}$ is the coagulation coefficient between size bin 'i' and 'j' and N$_j$ is the
PNC in size bin 'j'. Here 'i' was represented by 8 nm diameter and 'j' was varied between 25 and 900 nm
diameter. Following Seinfeld and Pandis, 2008, the coagulation coefficient was calculated as:

$$K_{ij} = \frac{2kT}{3\mu} \frac{(D_{pi} + D_{pj})^2}{(D_{pi} \times D_{pj})}$$


Here 'Dp' is the particle diameter, $k$ is the Boltzmann constant (1.38 x 10$^{-19}$ cm$^2$ kg s$^{-1}$), μ is the viscosity of air
(1.81 x 10$^{-7}$ kg cm$^{-1}$) and T is the surface temperature in Kelvin. Next, the condensation sink was calculated
based on the assumption that the properties of condensable vapors were similar to sulfuric acid, an important
condensable gas for the condensational growth of nucleated particles (Kulmala et al., 2013). CS is defined as:

$$CS = 2\pi . D_j . \sum_j \beta_{mj} . d_j . N_j$$

where D is the diffusion coefficient for sulfuric acid ~0.117 cm$^{-2}$s$^{-1}$ (Gong et al., 2010) and $\beta_{mj}$ is the size
dependent transition correction factor. The number concentration of aerosol particles in the size range of 5-25
nm was used to estimate the FR and GR. The CS and F$_{coag}$ were calculated using the entire PSND (~5-900 nm).
The units of FR, GR, CS and F$_{coag}$ are cm$^{-3}$s$^{-1}$, nm h$^{-1}$, s$^{-1}$ and cm$^{-3}$s$^{-1}$ respectively.

**3.2 NanoMap**
The NanoMap method developed by Kristensson et al.(2014), aims at representing the spatial distribution of
regional NPF events based on the meteorological backward trajectories and continuous PNSD measurements at
a point station. NanoMap gives an estimation of where NPF takes place at the point of formation of 1.5 nm
diameter particles up to at least 500 km distance upwind of the sampling site. This method is based on the
assumption that the measured NPF is a regional event and takes place over the full area covered by the regional
NPF event with the same nucleation start and end time. The NanoMap procedure follows four basic steps - (i)
classification and identification of type I NPF events (ii) choosing the start and end time of particle formation at
the lowest bin size (iii) determining the end of growth time i.e. the growth of newly formed particles can no
longer be followed in size distribution spectrum and (iv) plotting of a geographical position of the NPF events
based on the meteorological backward trajectories and selection made in steps i) to iii). The meteorological
backward trajectories have been calculated using the HYSPLIT model and meteorology data input from 1º
resolution Global Data Assimilation System (GDAS, Draxler and Rolph, 2003).

NanoMap is intended both for short data set (months to a few years) and longer data sets (several years or
longer, where even the probability of formation as a function of geographical region can be estimated). In this
study, the dataset of four months (from November 2016 to February 2017) is used to explore where the events
take place. The lowest detection limit diameter of WRAS is 5 nm and nucleated particles need some time to
grow from the initial size of 1.5 nm (Kulmala et al., 2013). The time shift between the real time formation and
observed time formation of particles was set to 1.5h based on the reported mean GR (~2.58 nm h$^{-1}$) at the
sampling site. Uncertainties related to event classification, selection of starting and ending times, simultaneous



event assumption and trajectory uncertainty are not negligible for the NanoMap method (Kristensson et al.,
198   2014).


**4. Results and Discussion**
The PNSD data measured during the study period were analyzed to identify the NPF events at HACPL. NPF
events occurred on ~40% of the measurement days (47 days out of 115 days). Of the total 47 NPF events, 13
events were identified as strong events and 34 events as weak events. The strong and weak events were
classified based on the number concentration of nucleation mode particles during the event. The events with
$N_{nuc}$ concentration greater than $6.0*10^3$ $cm^{-3}$ were referred to as strong NPF events and vice-versa. Out of these
47 NPF events, a strong event day (12[th] December 2016) was selected for a more detailed analysis. The
sampling site was under a varying influence of regional pollution throughout the study period (Mukherjee et al.,
2018). The characteristics of NPF were studied in terms of growth rate (GR), condensation sink (CS), formation
rate (FR) and coagulation ($F_{coag}$). The characteristics of the NPF day were compared with the non-NPF day -
14[th] December 2016. The role of meteorology and aerosol chemical composition on NPF was also evaluated.
Further, the contribution of newly formed particles to CCN was evaluated.

**4.1 Characteristics of NPF and non-NPF Event**
Figure 1 gives the colorplot of size distribution of aerosol particles on 12[th] and 14[th] December. In general, the
formation of new particles is governed by (i) precursor gases (ii) meteorological conditions like solar radiation,
temperature, relative humidity and wind speed and (iii) condensation sink. Figure 2 gives the diurnal variation
of different meteorological parameters on the NPF and non-NPF day. During the NPF day, there was a sharp
decrease in PNC from $~5.8*10^3$ $cm^{-3}$ to $4.1*10^3$ $cm^{-3}$ at 08:00 hrs. This decrease was related to an increase in the
boundary layer height. The gradual increase in boundary layer height is believed to bring cleaner air from aloft
to the ground, thereby lowering the CS to about $1.8*10^{-2}$ $s^{-1}$ a few hours later. Under low CS conditions, a
sudden enhancement in the concentration of $N_{nuc}$ ($~1.3*10^4$ $cm^{-3}$) was observed at 11:00 hrs. The formation rate
($J_5$) of particles was calculated as $~6.6$ $cm^{-3}$ $s^{-1}$. The CS is a measure of the amount of surface of the pre-existing
aerosol particles available for the semi-volatile gases to condense on to. Thus the low aerosol surface favoured
accumulation of condensable species in the gas phase, which nucleated and grew the new particles. The
formation of new particles was observed approximately 60-90 minutes later than the onset of photo-oxidation.
This delay was likely due to the lower limit of WRAS (5.14 nm) where it takes some time for nucleated cluster
particles at 1.5 nm diameter to grow to the detectable size at 5.14 nm). This is also the reason why the
representative value of CS in Table 1 at the onset of nucleation is chosen as one hour prior to the first observed
nucleation by the WRAS at 5.14 nm. During the nucleation process, high solar radiation (mean $~437\pm31$ $Wm^{-2}$),
low relative humidity (mean $~28\pm3$ %) and low wind speed (mean $~3\pm0.7$ $ms^{-1}$) were recorded. This hot and
less humid condition tends to favour the enhancement of atmospheric nucleation (Hamed et al., 2011).
According to Figure 1, the particles were continuously formed at 5 nm diameter for a minimum of 2 hours and
then grew at a rate of 2.5 nm $hr^{-1}$ (12[th] Dec) to a median size of 40 nm in due course of approximately 6 hours.
The growth of particles shifted in the particle size distribution from nucleation to an Aitken ($N_{ait}$) and/or
accumulation ($N_{acc}$) regime in the early morning of 13[th] December (01:00 hrs). Similar type of nucleation events
have been observed by other researchers (Dal Maso et al., 2005; Kulmala et al., 2004; Wu et al., 2007; Wang et



al., 2013; Kanawade et al., 2014; Leena et al., 2016; de Espana et al., 2017; Li et al., 2017). The CS increased
gradually during the event with its hourly averaged maximum ($4.9*10^{-2}$ $s^{-1}$) around 18:00 hrs, demonstrating
that the photo-oxidation has caused the rapid production of condensable vapors (Figure S1). The diurnal
variation of coagulation sink with its peak (hourly maximum ~ 9 $cm^{-3}$ $s^{-1}$) at 13:00 hrs suggested the maximum
coagulation removal of newly formed particles during this time period (Figure S1). The coagulation sink for
other strong NPF days is reported as an average throughout the day ± standard deviation in Table 1.

On the other hand, the non-NPF day (14th December) tended to have particle size distribution without a peak in
$N_{nuc}$ and contained only $N_{ait}$ and/or $N_{acc}$ particles. The mean PNC during 11:00-16:00 hours was observed to be
$6.6*10^3 \pm 1.5*10^3$ $cm^{-3}$ with the GMD calculated for the entire size distribution, varying between 45 to 80 nm.
During this period, high relative humidity (mean ~$86 \pm 4$ %), high wind speed (mean ~$6 \pm 1$ $ms^{-1}$) and low solar
radiation (mean ~$238 \pm 62$ $Wm^{-2}$) was recorded. It is believed that high relative humidity (>80%) tends to limit
the availability of gaseous sulphuric acid (a key precursor for NPF formation) in ambient air and thus inhibits
the formation of new particles (Hamed et al., 2011).

The ambition was to calculate NPF parameters only during days with strong NPF. From the 13 strong NPF
event days, 3 of these had gaps in data for a few hours, and NPF parameters were not possible to calculate, and
hence only 10 strong events have been analyzed for NPF parameters. For these 10 events, the growth rate of
nucleation mode particles was found to be linear with an average growth rate of $2.6 \pm 0.4$ nm $hr^{-1}$. The CS on
NPF days was much lower than on non-NPF days (~$4.2$ - $4.4*10^{-2}$ $s^{-1}$). The characteristic parameters for all
strong NPF events days and 2 non-NPF days (for reference) are summarized in Table 1.

**4.2 Role of aerosol chemical composition**

Figure 3 gives the diurnal variation of $N_{nuc}$ concentration, volume concentration of Aitken mode particles and
mass concentration of aerosol species for the strong NPF day during December 12, 2016. The role of these
parameters during the nucleation process was studied by dividing the activity period into 3-time windows. The
first window represents 09:30 to 11:00 hrs, which includes the formation of new particles. The window no. 2
from 11:00 to 13:30 hrs characterizes the increase in newly formed particles, and window no. 3 (13:30 to 18:00
hrs) represents the growth stage with an increase in volume concentration of Aitken mode particles and a
decrease in $N_{nuc}$ concentration. During the first time window, a considerable decrease in aerosol mass
concentration coincided with the rising boundary layer height, which led to the dilution of pollutants. The $N_{nuc}$
concentration and Aitken volume concentration were at a minimum during this window. Since the formation of
new particles is governed by the nature of pre-existing aerosol particles, the particle acidity was evaluated by
following the methodology of Engelhardt et al., 2013. The ratio of ammonium to anions (sulfate and nitrate) was
calculated as 1.08, which indicated the neutralized nature of aerosols. The neutralized aerosols make the
atmospheric conditions favourable for NPF. On the other hand, the aerosols on non-NPF day (14th December)
were found to be slightly acidic in nature with the ratio of ammonium to anions varying between 0.88 and 0.90.
Therefore it is assumed that the ammonium available in atmosphere would be used up by the pre-existing anions
for their neutralization and thereby could potentially suppress the NPF (Pikridis et al., 2014). During window
no. 2, a significant increase in the $N_{nuc}$ concentration was observed. In the third time window, the gradual




increase in Aitken volume concentration and decrease in $N_{nuc}$ concentration was observed. This was
accompanied by the gradual increase in mass concentration of aerosol (organics, sulfate, nitrate and
ammonium), but with a time lag of ~2 hours. The ACSM is not able to detect the smallest particles in the Aitken
mode range, but only after they have grown to detectable sizes a few hours later. The organic components -
OOA, HOA, BBOA all increase during the afternoon and evening and were identified on the basis of
fragmentation patterns of the ACSM and source/receptor analysis of high resolution mass spectra using Positive
Matrix Factorization (PMF) model (Paatero and Tapper, 1994). More details on PMF methodology used in this
study are given in our earlier work (Mukherjee et al., 2018). OOA is the oxygenated organic aerosol and relates
to secondary organic aerosol. HOA and BBOA refer to hydrocarbon-like organic aerosol and biomass burning
organic aerosol. HOA and BBOA are the surrogates of primary organic aerosol. The gradual increase in aerosol
mass concentration suggested that (i) the particle size has grown large enough ($\geq$ 40 nm) to be able to be
detectable by the ACSM as explained earlier and (ii) that there is formation of secondary aerosol components
during the growth of newly formed particles. It is believed that the production of low volatile organic vapours
and nitrate have caused these new particles to grow quickly, thereby enhancing the CCN concentrations
significantly (Figure 3e). Similar features have also been reported by previous NPF studies in diverse
environments (Kulmala et al., 2004; Zhang et al., 2004; Crilley et al., 2014; Li et al., 2017).

## 4.3 Contribution of NPF to CCN

In order to become CCN, the freshly nucleated particles must attain a minimum size of ~40nm. To determine
the probable contribution of these particles to CCN, simultaneous measurements of CCN were carried out.
Figure 3(e) gives the hourly variation of CCN concentration at five supersaturation levels on the NPF day. The
CCN concentration increased gradually coinciding with the (i) increase in $N_{ait}$ concentration, and (ii) increase in
volume concentration of Aitken mode particles. This increase in CCN concentration was attributed to the
increase in number concentration of particles fulfilling the size criteria for CCN formation. The probable
contribution of newly formed particles to CCN formation was evaluated in terms of percent increase in CCN.
The percent increase was calculated by comparing the CCN concentrations before and after the nucleation
event. The two time windows were selected as - "Window 1 ($W_1$) and Window 2 ($W_2$)" representing the time
periods before and after the nucleation event respectively (Please see Figure 1). While choosing these windows,
the care was taken to ensure that: (i) there were no primary emissions, (ii) meteorological conditions were
sufficiently stable, and (iii) the geometric mean diameter (GMD) of particles was well above 40 nm. In this
study, primary organic aerosol (POA= HOA+BBOA) was used as a marker for the primary emissions. The
stable meteorological conditions were defined by stable wind speed and wind direction. Further it should be
noted that for the days with successive NPF events, $W_1$ was chosen just before the initiation of nucleation event.
The second time window was selected when the GMD of aerosol particles was well above the 40 nm. Thus the
change in CCN concentration was calculated as:

$$\Delta CCN = \frac{CCN_{W_2} - CCN_{W_1}}{CCN_{W_1}}$$




Table 2 gives the change in CCN concentration and other parameters for the 13 strong NPF events, which is
~27% of all the events. These days are chosen on the basis of availability of all the simultaneous measurements
(aerosol chemical composition, PNC and meteorological parameters) and the fulfillment of the requirements for
the number concentration of strong events in section 4. The CCN measurements were selected at 0.52% SS in
this study, which would correspond to activation diameters around 70 nm for hygroscopic particles (Singla et
al., 2017). A considerable increase in CCN concentrations was observed after the event on each day with an
average increment of $42 \pm 27\%$ (Table 2). In this study, POA (primary organic aerosol) was also used as a
surrogate for BC (tracer for primary emission). The parameters - change in activated fraction and change in
CCN/POA, were calculated to identify the source of increased CCN concentrations. If the change in activated
fraction is similar to the change in CCN/POA, then primary emissions are the likely contributors for increased
CCN concentrations. On the other hand, if the change in activated fraction differs from the change in
CCN/POA, newly formed particles are the major contributors to increased CCN concentrations (de España et
al., 2017). It is observed from Table 2 that the change in activated fraction differs significantly from the change
in CCN/POA for each day. As an example, the metrics for $12^{th}$ December shows that, the change in CCN/CN is
26% and the change in CCN/POA is -18%. This significant difference (between W1 and W2, refer figure 1)
indicated new particle formation as the important source of CCN at our site. The significant difference between
$\Delta$(CCN/CN) and $\Delta$(CCN/POA) for all the chosen events suggested NPF as the distinct source of increased
CCN. Further, $\Delta$(CCN/NR-PM$_1$) was also calculated for accounting the BLH effect, if any. The positive change
of $\Delta$CCN/NR-PM$_1$ indicated that higher CCN/NR-PM$_1$ after the event was the result of NPF and not the dilution
of BLH.

**4.4 Cluster analysis**
The cluster analysis was carried out to outline the relationship of air masses with the observed particle number
size distribution of aerosols. The trajectories were simulated by using the Hybrid Single-Particle Lagrangian
Integrated Trajectory (HYSPLIT) model (Draxler and Rolph, 2003). The 120 hour backward trajectories were
calculated for every hour terminating at a height of 100 m above the ground level. In total, 2760 trajectories
were calculated during the study period. The transport pathways of the observed air masses were grouped into
five clusters on the basis of total spatial variance (TSV). Each cluster congregated into different geographical
origin(s). The air-mass backward trajectory calculations and cluster analysis were performed by using Zefir
(Igor-based software developed by Petit et al., 2017). Before running the cluster analysis, backward trajectory
data was down-weighted by using the discrete weighting function in Zefir (Petit et al., 2017).

Figure 4(a) shows the derived fire hotspots (good indicators for biomass burning activities), from MODIS
(Moderate Resolution Imaging Spectro-radiometer) and geographical origin of each cluster and Figure 4(b)
shows the corresponding PNSD observed at the receptor site. The PNSD of each cluster was constructed by
averaging the particle number size distribution for each time step belonging to that particular cluster. It was
observed that air masses arriving at the receptor site generally originated from the inland continent and
represented 95% of the total back trajectories (cluster C1 - 51%, cluster C2 - 13% and cluster C3 - 25%). Cluster
C1 originated from Central India, C2 from northwest India and C3 from southeast India. On the other hand,
clusters C4 and C5 began from Iran and Saudi Arabia respectively and contributed only 4% and 7% respectively



to the total back trajectories. Before reaching the receptor site, cluster C4 traveled long through the land and
cluster C5 traveled both through the sea and land. Both the clusters - C4 and C5 were identified as fast moving
clusters. However, only cluster C4 was found to be a relatively cleaner air mass as the total NR-PM$_1$
concentration for C4 was found to be 7.2 µgm$^{-3}$. On the other hand, continental originated clusters were
identified as slow moving clusters and were associated with higher PM$_1$ concentration (C1 ~15 µgm$^{-3}$, C2 ~11
µgm$^{-3}$ and C3 ~13 µgm$^{-3}$). The results of PNSD showed that the formation and growth of nucleation mode
particles were significant for clusters associated with continental air masses (Figure 4). Clusters (C4 and C5) did
not contain any nucleation although the mean total particle number concentration was comparable. The
characteristics of aerosol particles associated with each cluster were analyzed by dividing the PNC into 3
distinct modes - nucleation (5-25 nm), Aitken (25-100 nm) and accumulation (100-1000 nm) mode. The diurnal
variation of PNC in these 3 modes for each cluster is shown in Figure S2.
Nucleation mode - The particles in this mode showed significant increase in number concentration from 11:00
to 15:00 hours for clusters C1 and C2 with the considerable decrease in N$_{ait}$ concentration. This increase was
mainly influenced by the formation and growth of new particles under favourable meteorological conditions.
However, a faded increase in the number concentration was also observed for cluster C3 between 13:00 and
15:00 hours ~1155 cm$^{-3}$). This diminished nucleation was attributed to the interference caused by a sudden
increase in N$_{ait}$ concentration at the onset of nucleation. Thereafter, the nucleation event persisted for few hours.
Conversely, no nucleation process was observed for clusters C4 and C5. The diurnal variation of N$_{nuc}$ particles
exhibited high concentration during morning, afternoon and evening hours. However, these N$_{nuc}$ peaks
coincided well with the N$_{ait}$ concentration peaks. The simultaneous peaks in the two modes suggested the
emission of nucleation particles directly by local sources.
Aitken mode - For all the air mass cluster types, the particles in this mode reflected a morning (~08:00 hrs) and
an evening (~18:00 hrs) peak corresponding to traffic rush hours. In addition, clusters C3, C4 and C5 showed
somewhat elevated concentrations during afternoon and early evening in the PNC. These peaks may be
attributed to the fresh anthropogenic emissions by cooking related sources (Mukherjee et al., 2018). The mean
PNC during the morning peak was 2.5*10$^3$ cm$^{-3}$, 2.6*10$^3$ cm$^{-3}$, 2.4*10$^3$ cm$^{-3}$, 5.5*10$^3$ cm$^{-3}$ and 5.2*10$^3$ cm$^{-3}$ for
clusters C1, C2, C3, C4 and C5 respectively. In addition, a significant noon peak related to cooking was
observed for cluster C3 (~ 2663 cm$^{-3}$).
Accumulation mode - The PNC in this mode showed a stable pattern as a function of time for clusters C1, C2
and C3. This stable pattern of PNC indicated the regional transport of anthropogenic pollution to the receptor
site. The regional transport of organics and sulfate has been reported during the same period in our earlier work
(Mukherjee et al., 2018). The 24-hour average concentration of N$_{acc}$ particles was measured as 1.8*10$^3$ cm$^{-3}$,
2.0*10$^3$ cm$^{-3}$, 1.8*10$^3$ cm$^{-3}$, 1.4*10$^3$ cm$^{-3}$ and 2.1*10$^3$ cm$^{-3}$ for clusters C1, C2, C3, C4 and C5 respectively.
The above analysis revealed that the clusters associated with continental air masses favoured new particle
formation under the influence of long range transported anthropogenic pollution to the sampling site. However,
the clusters associated with foreign air masses did not favour new particle formation. Based on the diurnal




variation of $N_{nuc}$, $N_{ait}$ and $N_{acc}$, it was assumed that the concentration of $N_{ait}$ is influencing the occurence of NPF.
It was noticed that the concentration of $N_{ait}$ during the morning hours was well below ~$3*10^3$ cm$^{-3}$ for
continental clusters and much above ~$5.5*10^3$ cm$^{-3}$ for the foreign clusters. Also, the nucleation process was
found to be inhibited for cluster C3 by the sudden increase in $N_{ait}$ concentration (~$2.7*10^3$ cm$^{-3}$) during the
afternoon hours. These statistics suggest that the number concentration of ~$3*10^3$ cm$^{-3}$ may be acting as a
threshold value of $N_{ait}$ particles for the appearance of NPF. The concentration of $N_{ait}$ below ~$3*10^3$ cm$^{-3}$ was
found to favour NPF at this site during the study period and vice versa. On the other hand, clusters 4 and 5 also
show a decrease in $N_{ait}$ after the morning period, and still NPF does not take place. This may be indicative of
low addition of condensable vapours needed for NPF for the cluster 4 and 5 air masses, whereas it is likely that
the addition of condensable vapours for NPF is high enough in the Clusters 1-3. The insight into the relationship
between $N_{ait}$ concentration and nucleation process was further comprehended by analyzing the individual days.
As an example, three days - 12$^{th}$ Dec 2016, 19$^{th}$ Dec 2016 and 19$^{th}$ Nov 2016 were chosen as representative days
for NPF, non NPF and weak NPF respectively. The selected days belongs to cluster C1. The cluster C1 was
chosen because it contributed most (~51%) to the total back trajectories.

Figure 5 shows the diurnal variation of PNC in 3 modes ($N_{nuc}$, $N_{ait}$ and $N_{acc}$) for three selected days. The
variation in meteorology - temperature (T), relative humidity (RH) and wind speed (WS) is shown in Figure S3.
The PNC in the Aitken mode during 07:00 - 09:00 hrs was found to be ~$2.2*10^3$ cm$^{-3}$, ~$3.6*10^3$ cm$^{-3}$ and
~$2.6*10^3$ cm$^{-3}$ on NPF, non-NPF and weak NPF day respectively for the three selected days. The variation of
solar radiation and temperature was found to be similar on all the days. On the other hand, RH varied
considerably. During the NPF day, the mean RH was ~$36 \pm 7$ % with the minimum of ~23% during the
nucleation process. The non-NPF and weak NPF day showed a mean RH of ~$57 \pm 4$% and ~$26 \pm 7$%
respectively. The magnitude of WS was also variable. The NPF day, non-NPF day and weak NPF day showed
mean WS of $6.2 \pm 2.8$ ms$^{-1}$, $2.4 \pm 1.2$ ms$^{-1}$ and $4.1 \pm 1.6$ ms$^{-1}$. Further, high WS was recorded just before the
nucleation event (08:00 to 09:00 hrs), ~8.7 ms$^{-1}$ on the NPF day, ~1.7 ms$^{-1}$ on the non-NPF day and 6.8 ms$^{-1}$
(10:00 hrs) on the weak NPF day. On the basis of these statistics, it was concluded that stable, relatively low
background Aitken mode concentrations with low RH favoured the new particle formation on 12$^{th}$ December
under high wind speed, temperature and solar radiation conditions. On the other hand, fresh anthropogenic
emissions and unfavourable meteorological conditions inhibited the nucleation activity on 19$^{th}$ December. The
fresh anthropogenic emissions were observed during morning (~$3.4*10^3$ cm$^{-3}$) and noon (~$3.1*10^3$ cm$^{-3}$) on 19$^{th}$
November as well. However, the $N_{ait}$ concentration was much lower than that observed on the non-NPF day.
Secondly, the molar ratio of ammonium to inorganic anions was calculated as ~1.4. Since both source
(ammonia) and sink (pre-existing particles due to fresh anthropogenic emissions) entities were present in high
amount, it was not likely inhibiting NPF as suggested in section 4.2. Instead, relatively unfavourable
meteorological conditions inhibited strong formation on the weak NPF event day. Moreover, the interference by
fresh emissions during noon time was assumed to suppress the nucleation activity and hence lead to the weak
NPF day with maximum $N_{nuc}$ ~ $4.5*10^3$ cm$^{-3}$ at 15:00 hrs.

Since the nucleation process in this study was influenced by the transported anthropogenic pollution, the
chemical composition of each cluster was also evaluated. Table 3 gives the mass concentration of organic and



inorganic components of aerosol particles. The comparable mass concentration of nitrate and HOA indicated no
influence of long range transport on NPF. The mass concentration of sulfate, ammonium, BBOA and OOA was
higher for the continental clusters 1-3 as compared to the foreign clusters 4-5. This difference may be attributed
to the travel pathways of each cluster. Clusters C4 and C5 were identified as fast moving clusters and had a
lower travel time over the continent. Therefore, it was assumed that high mass concentration of sulfate,
ammonium, BBOA and OOA were indicative of local or regional emissions. The continental clusters originated
from the biomass burning affected areas and identified to represent regional, or in other words, long range
transported aerosol particles (Figure 4). The higher mass concentration of BBOA was attributed to the regional
transport of biomass burning related aerosols. With a reported lifetime of $3.8 \pm 0.8$ days (Edwards et al., 2006),
biomass burning plumes are often transported for thousands of kilometers (Anderson et al., 1996; Andreae et al.,
1988). The higher mass concentration of sulfate was also related to long range transport. Since no direct
measurements of $SO_2$ were available at this site, the spatial distribution of $SO_2$ emissions over the Indian region
was extracted from the OMI satellite (Figure S4). Figure S4 shows high $SO_2$ emissions in central India. The
biomass burning events and power plants seems to serve as the two major sources of $SO_2$ emissions during the
study period (Figure 4 and S4). It is reported that $NO_2$ can act as an important oxidant for the conversion of $SO_2$
to sulfate under biomass burning influenced anthropogenic pollution (Xie et al., 2015). Further, the continental
clusters also displayed higher mass concentration of OOA. This high concentration was attributed to the
atmospheric changes in biomass burning aerosols often resulting in the enhanced fraction of OOA and
degradation of biomass burning related species (Cubison et al., 2011; DeCarlo et al., 2008; Yokelson et al.,
2009). Therefore, it can be concluded that the formation of new particles took place under the influence of
biomass burning affected anthropogenic pollution. The chamber study by Henningan et al., 2012 has shown that
exposing the biomass burning plume to UV light initiates photo-oxidation thereby creating a strong nucleation
burst. The field studies have also reported the nucleation activity in fresh (Hobbs et al., 2003) and aged fire
plumes (Andreae et al., 2001; Wu et al., 2016).

**4.5 NanoMap**
The NanoMap method (Kristensson et al., 2014) was used to identify the spatial distribution of NPF events
during the formation of 1.5 nm particle diameter. It is only applicable to the areas which are upwind and up to
500 km from the sampling site. This method helps in determining the spatial NPF events occurring at the same
time as at the sampling site. In this study, the dataset of 4 months (November 2016 to February 2017) was used
to earmark the source area and frequency of the formation of 1.5 nm diameter particles along the trajectories.
The data was available for 95% of total days (115 days out of 120 days). During these days, 34% of days (~40
days) were identified as type-I NPF event days. All the events were easily analyzable with respect to start time
of NPF and end of growth time.

Figure 6 shows the location of formation of 1.5 nm diameter upwind of the HACPL site with a grid resolution of
0.1 x 0.2°. A large NPF frequency was observed in the areas favoured solely by the continental air masses. The
biomass burning mixed anthropogenic plume from the Central India seems to favour the occurrence of the NPF
events upwind of HACPL. However, no NPF events were registered with the air masses coming from other
directions. The probable reason could be the non-availability of sufficient precursors ($SO_2$ or $H_2SO_4$) at the



sampling site. Figure S4 shows the $SO_2$ emission for the time period - November 2016 to February 2017. It has
been reported that $SO_2$ once emitted, readily reacts with hydroxyl radical in the atmosphere to produce $SO_3$. The
$SO_3$ formed then reacts quickly with water vapour to produce sulphuric acid or depending on the meteorological
conditions and availability of oxidizing substances; $SO_2$ may be transported hundreds of kilometres before it
forms sulfuric acid (Erduran and Tuncel, 2001). Another possibility is that the oxidation of trace gases emitted
from biomass burning produced low-volatile condensable vapours, which nucleated in the biomass burning
plume (Wu et al., 2016). Moreover, a recent study found that biomass burning can enhance the conversion of
$NO_2$ to HONO, which is one of the main sources of OH (Hobbs et al., 2003; Yokelson et al., 2009; Nie et al.,
2015). Therefore the biomass burning mixed anthropogenic plume was expected to transport ample amount of
precursor gases ($SO_2$ or $H_2SO_4$) initiating the nucleation upwind of the HACPL site. The spatial extension of the
NPF of 1.5 nm diameter particles extended up to several hundred kilometres from HACPL according to the
NanoMap method. In reality, the extension could be even higher than this. This is not possible to observe due to
sudden interruptions in the GR of newly formed particles during NPF, and the limitations of the NanoMap
method (Kristensson et al., 2014).

**Conclusion**
The simultaneous measurements of particle number size distribution, aerosol chemical composition and
meteorology were performed at Mahabaleshwar from November 2016 to February 2017. The data was analyzed
to identify the occurrence of nucleation events. Ample NPF events were observed with a frequency of ~40%.
Certain meteorological conditions were favourable during the study period for NPF, and local and long range
transported aerosol sources played a significant role in the occurrence of NPF. The main conclusions drawn
from this study are listed below:
(i) All the NPF events began around 10:00-11:00 hours. Ten of the strongest NPF events observed had an
average growth rate, formation rate, condensation sink and coagulation of $2.6 \pm 0.4$ nm h$^{-1}$, $2.8 \pm 1.4$ cm$^{-3}$ s$^{-1}$, 2.2
$\pm 2.9 *10^{-2}$ s$^{-1}$ and $1.6 \pm 1.0$ cm$^{-3}$ s$^{-1}$ respectively.
(ii) Fresh anthropogenic emissions (resulting in high $N_{ait}$ concentration) or unfavourable meteorology led to
weak or no NPF events. On the basis of cluster analysis of backward trajectories, a concentration lower than
$3*10^3$ cm$^{-3}$ was earmarked as the threshold value of Aitken mode particles favoring nucleation. During the non-
NPF event days $N_{ait}$ concentrations were significantly higher.
(iii) The analysis suggested that (a) the air masses influenced by biomass burning fromnorth-east and (b) high
wind speed just before the nucleation event favored nucleation at HACPL.
(iv) The growth of freshly nucleated particles persisted for ~ 6-7 hours and led to the significant enhancement in
mass concentration of aerosol (OOA, sulfate and nitrate).
(v) The NPF events acted as the significant source of CCN with the mean percentage increment of ~$53 \pm 36\%$.
(vi) NPF took place up to several hundred kilometers upwind to the north-east of HACPL.

**Data availability**
The data used in this study are from the data repository of HACPL, part of IITM, Pune and will be made
available on request.




**Acknowledgments**
Authors are grateful to all the team members of High AltitudeCloud Physics Laboratory (HACPL) of IITM.
HACPL is fully funded byMinistry of Earth Sciences (MoES), Government of India, NewDelhi. The data used
in this study are from the data repository of HACPL, part of IITM, Pune. Authors are thankful to the Director of
IITM for his support and encouragement. Vyoma Singla extends special thanks to DST, SERB for NPDF
fellowship (Fellowship number: PDF/2017/002428) and Director, IITM for providing all the facilities. Swedish
research council FORMAS (project no. 2014-951) is acknowledged for support of the salary for Kristensson.

**Competing interests**
The authors declare that they have no conflict of interest.

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



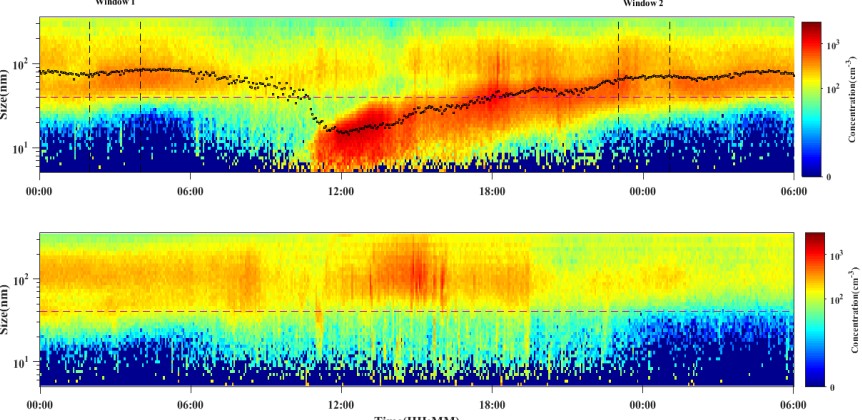

757

Figure 1: The spectrum of the particle number size distribution on a strong NPF day - 12th December 2016
(upper panel) and a non-NPF day - 14th December 2016 (lower panel). Window 1 and Window 2 represents the
stable atmospheric conditions before and after the nucleation started, respectively. The black dotted line in the
upper panel represents the geometric mean diameter of particles. The horizontal dashed line in both the panels
represents the diameter of particles at 40 nm.

763

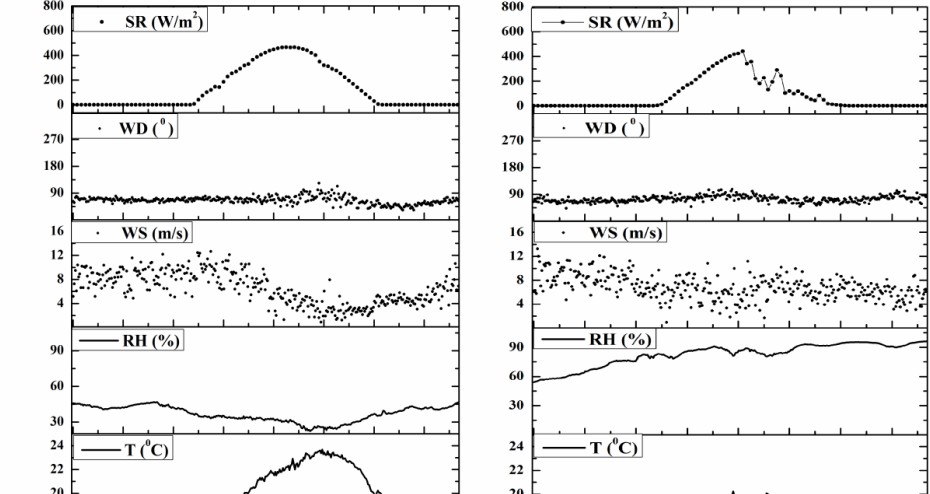

764

Figure 2: Diurnal variation of meteorological conditions - Solar radiation (SR), Wind direction (WD), Wind
speed (WS), Relative Humidity (RH) and Temperature (T) on the special case NPF day (left panel, December
12, 2016) and the non-NPF day (right panel, December 14, 2016).

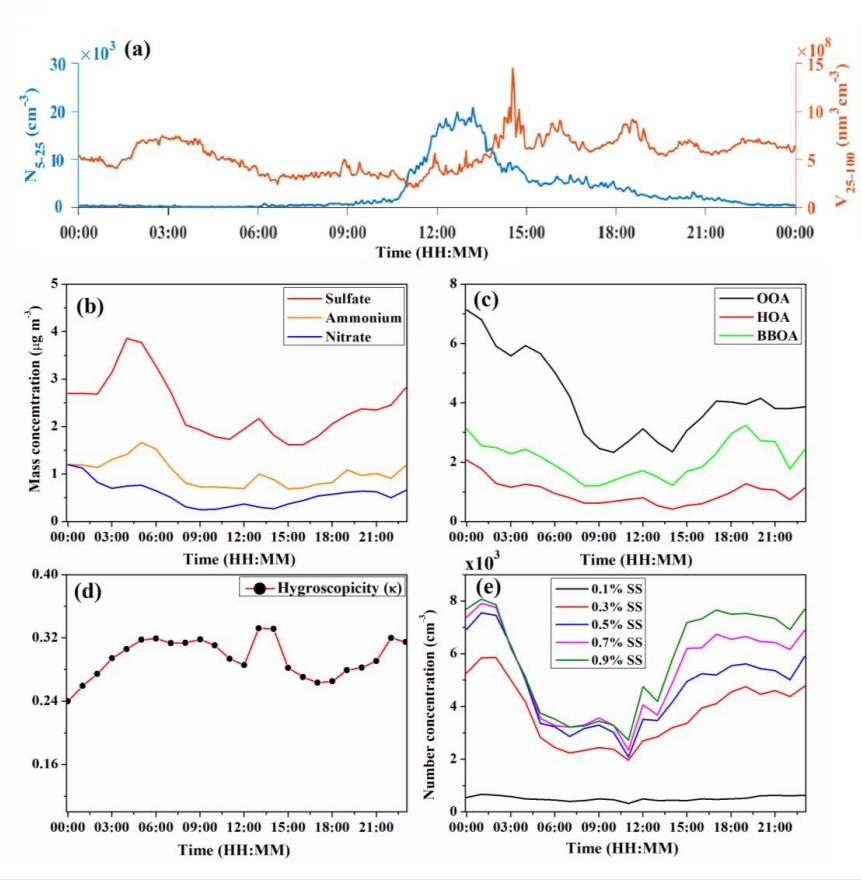

768

Figure 3: Particle properties during the strong NPF event day on December 12, 2016. Diurnal Variation of (a) nucleation mode number concentration and aitken mode volume concentration (b) mass concentration of secondary inorganic aerosol species (c) mass concentration of secondary organic aerosol species (OOA = oxygenated organic aerosol, HOA = hydrogenated organic aerosol, BBOA = biomass burning organic aerosol) (d) calculated hygroscopicity and (e) CCN number concentration at five supersaturations - 0.1%, 0.3%, 0.5%, 0.7% and 0.9%.










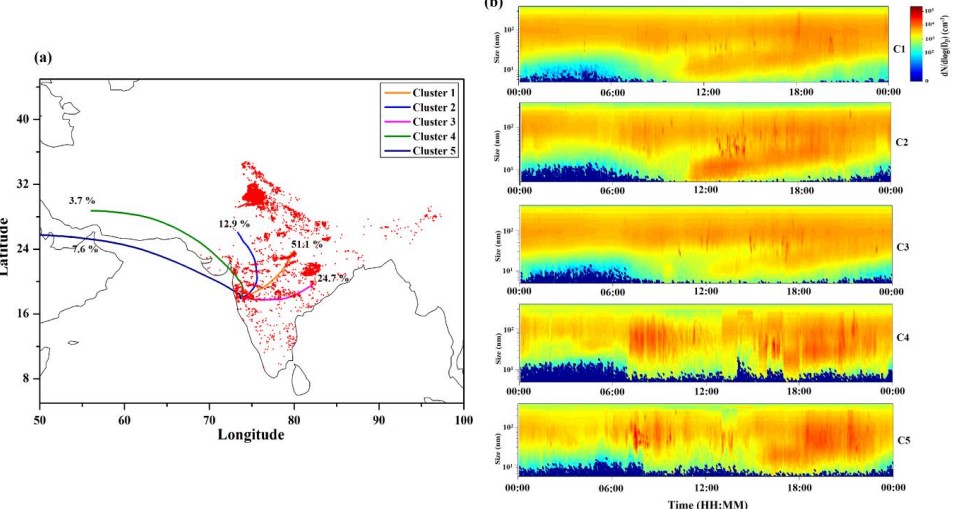


Figure 4: Results of 120 h air mass backward trajectories at 100m above ground level during the study period
(November 2016 to February 2017). Left Panel (a) - The mean trajectories of each cluster obtained through
cluster analysis. Red dots represent the fire hotspots as identified by MODIS. Right Panel (b) - The spectrum of
average particle number size distribution corresponding to each mean cluster.

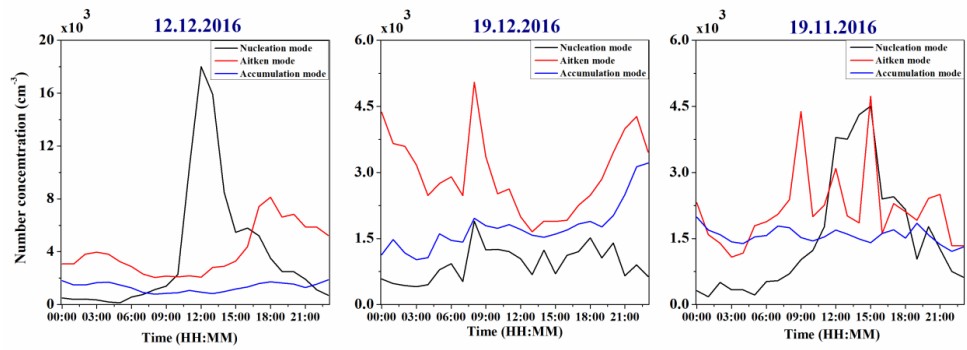


Figure 5: Diurnal Variation of particle number concentration in three modes - nucleation, aitken and
accumulation mode for a NPF day -12th December 2016, a non-NPF day - 19th December 2016 and a weak-NPF
day - 19th November 2016.





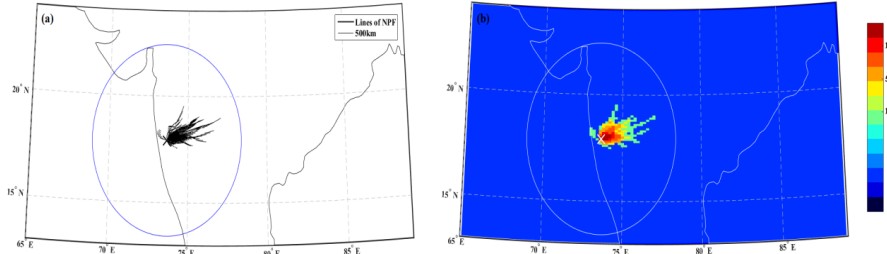

Figure 6: NanoMap results: Left Panel (a) - Black Lines represent the formation of 1.5 nm diameter particles.
Right Panel (b) - The frequency of NPF events at 1.5 nm diameter, with a grid resolution of 0.1 x 0.2 degrees for
latitude and longitude, respectively. The blue and white ovals limit the horizontal dimension of 500 km radius
from the sampling site.



Table 1: Summary of new particle formation events (strong event days) observed during November
2016 to February 2017. The start and end time were identified based on the change in GMD of
nucleation mode particles. The $F_{coag}$ and $J_5$ were calculated at the time of NPF formation. The CS
corresponds to the time period ~1 hour before the start of NPF at 5.14 nm diameter. The GMD of
nucleation mode particles (5-25 nm) was used to calculate the GR.

| Date | Start Time | End Time | $J_5$ (cm$^{-3}$ s$^{-1}$) | $F_{coag}$ (cm$^{-3}$ s$^{-1}$) | CS (* 10$^{-2}$ s$^{-1}$) | GR$_{5-25}$ (nm h$^{-1}$) |
|---|---|---|---|---|---|---|
| **NPF Days** | | | | | | |
| **8 Nov** | 11:00 | 18:00 | 4.9 | 3.4 | 2.1 | 2.3 |
| **9 Nov** | 10:00 | 17:00 | 6.6 | 5.2 | 2.0 | 2.4 |
| **10 Nov** | 11:00 | 17:00 | 6.9 | 5.4 | 2.2 | 2.4 |
| **12 Nov** | 11:00 | 17:00 | 3.9 | 3.0 | 2.3 | 2.5 |
| **18 Nov** | 10:00 | 16:00 | 7.3 | 6.3 | 2.2 | 3.3 |
| **2 Dec** | 10:00 | 17:00 | 2.0 | 1.5 | 2.1 | 2.2 |
| **12 Dec** | 11:00 | 18:00 | 6.6 | 5.1 | 2.0 | 2.5 |
| **13 Dec** | 10:00 | 16:00 | 5.2 | 4.4 | 2.9 | 2.3 |
| **17 Feb** | 12:00 | 18:00 | 2.4 | 1.8 | 2.1 | 3.0 |
| **18 Feb** | 11:00 | 18:00 | 4.1 | 3.3 | 2.5 | 2.9 |
| **Non-NPF Days** | | | | | | |
| **14 Dec** | - | | - | - | 4.4 | - |
| **15 Dec** | - | | - | - | 4.2 | - |



Table 2: Summary of NPF event days qualifying the criterion mentioned in section 4.3. POA here
refers to the sum of HOA and BBOA. The information on HOA and BBOA was obtained by using
Positive Matrix Factorization. All the parameters listed below are calculated at 0.52% SS.

| Date | ΔCCN | ΔCCN/CN | ΔCCN/POA | ΔCCN/NR-PM$_1$ |
|---|---|---|---|---|
| **07.11.16** | 40 % | 24 % | 61 % | 99 % |
| **08.11.16** | 29 % | 37 % | 65 % | 44 % |
| **09.11.16** | 43 % | 45 % | 77 % | 23 % |
| **10.11.16** | 19 % | 15 % | 32 % | 11 % |
| **18.11.16** | 21 % | 34 % | - 30 % | 25 % |
| **23.11.16** | 4 % | 29 % | - 20 % | 13 % |
| **11.12.16** | 75 % | 13 % | 2 % | 31 % |
| **12.12.16** | 93 % | 26 % | -18 % | 11 % |
| **13.12.16** | 33 % | 38 % | 58 % | 10 % |
| **18.12.16** | 5 % | 13 % | 8 % | 12 % |
| **03.01.17** | 51 % | 28 % | -12 % | 44 % |
| **24.01.17** | 76 % | 11 % | - 29 % | 8 % |
| **25.01.17** | 58 % | 24 % | - 31 % | 23 % |







Table 3: Summary of secondary aerosol components, both organics and inorganics. The mass
concentration is represented as mean ± standard deviation (µg m$^{-3}$) for the five clusters.

| Cluster | Sulfate | Nitrate | Ammonium | HOA | BBOA | OOA |
|---|---|---|---|---|---|---|
| C1 | 3.5 ± 3.1 | 1.0 ± 1.0 | 1.5 ± 1.2 | 0.9 ± 1.3 | 2.1 ± 2.4 | 4.0 ± 2.7 |
| C2 | 2.9 ± 1.7 | 0.9 ± 1.0 | 1.0 ± 0.8 | 0.7 ±1.3 | 1.6 ± 2.2 | 3.1 ± 2.1 |
| C3 | 3.4 ± 3.0 | 1.1 ± 0.9 | 1.5 ± 1.2 | 0.8 ± 1.1 | 1.5 ± 2.2 | 3.5 ± 2.6 |
| C4 | 1.2 ± 0.9 | 0.5 ± 0.4 | 0.6 ± 0.4 | 0.8 ± 1.8 | 1.3 ± 1.8 | 1.7 ± 1.1 |
| C5 | 1.4 ± 1.7 | 1.0 ± 1.1 | 1.1 ± 0.8 | 0.6 ± 0.9 | 1.3 ± 1.3 | 2.0 ± 1.4 |





