# Peer review of "New Particle Formation at a High Altitude Site in India: Impact of Fresh Emissions and Long Range Transport"

_Atmospheric Chemistry and Physics, 2018_

## Referee Comment (RC1) · Anonymous Referee #1 · 2 Aug 2018

The paper reports new particle formation (NPF) events observed at the High Altitude Cloud Physics Laboratory (HACPL) in Mahabaleshwar in Western India. They measured nucleation and growth rates as well as condensation and coagulation losses. The authors further analyze under what conditions NPF occurs by using a detailed case analysis and a cluster analysis. The paper concludes that NPF is favored by low relative humidity and when air masses reach the site from central India. Using the NanoMap method it is estimated that NPF events occur several hundred kilometers upwind of the site.

The paper presents an interesting data set on NPF. However, many claims and con-

clusions I could not see and follow, e.g. the determination of CCN to NPF (details see below). Furthermore, many conclusions are not well established and over-interpreted. The paper needs substantial improvements following the detailed comments below.

Line 218: the sharp decrease occurs between 4-6h according to Figure 3a and therefore before sunrise (Figure 2). This looks like a change of air mass as also seen in the change of hygroscopiciy. It would also be more helpful to present absolute humidity in Figure 2 to check for air mass changes. H

Line 225: sunrise it at 7.30h and NPF is seen at 11h. this is not 60-90 minutes later. Probably you mean estimated start of NPF.

Line 229ff: What time period do you take for the average values? By the way I do not consider 20-24°C as hot.

Line 233: With a GR of 2.5 nm/hr it is impossible to reach 40 nm in 6 hours.

Line 248: How do you know that sulfuric acid is driving nucleation? It could also be organics. Furthermore, it could also be other parameters like availability of nucleating species.

Line 195: this GR is for particles larger than 5 nm. However, for smaller particles GR is usually quite slower due to the Kelvin effect (see Kulmala 2013).

Line 256: The CS is only a factor 2 higher on non-NPF days. This is not a much higher value.

Line 263-264: 9:30 to 11:00h is the assumed time of formation of particles, while 11:00 to 13:30 h is the observed formation. Make this clear.

Line 266: The decrease in aerosol mass happens before. When NPF starts, PNC and aerosol mass are already low.

Line 269: What do you mean by "formation of new particles is governed by the nature of pre-existing aerosol particles"?

Line 270: Is this the equivalent ammonium to anion ratio? What time interval does this cover? Only after 9h?

Line 278: organics increase before inorganics. How do you interpret that?

Line 286: Much of the increase in aerosol mass is by HOA and BBOA and not due to additional mass from NPF. In Figure 1 it is seen that the concentration of particles 50-200nm increases after 14h. What can now be attributed to this increase and to particles grown in from NPF? What is the cut-off of the ACSM?

Line 289: this is speculation. CCN starts to increase already at 12h, one hour after observing NPF at 5 nm. How can these small particles be activated so early? Could it also be that particles in the 50-200 nm band start to grow due to condensation of organics and become CCN active?

Section 4.3: There are several issues with the estimation of NPF to CCN. The authors say that they compare time window 2 with window 1 and that GMD > 40 nm. In Figure 1 these criteria cannot be met but calculations are given. The notation given for the different parameters (CCN/POA, etc. . .) is unclear and not consistent. I had to consult the original publication to understand it. All values for delta(CCN/CN) are positive. I have a hard time to believe that the fraction of CCN is always larger shortly after nucleation (11:00-13:30h) than in the aged air mass when no nucleation mode CN are measured yet. This would mean that an extremely large fraction of nucleation mode particles (most particles are still <40 nm) activates. I also doubt that CCN/POA is a good measure for primary contributions to CCN. Along a trajectory POA decreases while CN may not change much. This pretends an increase in CCN fraction that is not real. As already mentioned above an increase in CCN may also be due to growth of 50-200nm particles. How do the authors account for this?

Section 4.4: this section contains many claims that can hardly be seen in the Figures and small variations in measurements are overly interpreted. Line 368: I do not see the decrease in Aitken mode concentration during increase of nucleation mode particles.

Line 371: Onset of nucleation is usually around 9:00h. I do not see a sudden increase in Aitken mode particles. In addition average Aitken mode concentration at that time is similar in cluster C2 and C3.

Line 372: It says: The diurnal variation of $N_{nuc}$ particles exhibited high concentration during morning, afternoon and evening hours". I do not see high concentrations throughout the day. I also do not see the simultaneous peaks of nucleation and Aitken modes. There are time shifts.

Line 378ff: It is claimed that Aitken mode has a morning and evening peak due to traffic. First of all some peaks are small and difficult to interpret. Second, there is no further proof that all this is local. Third, Figure 3 does not show any of these peaks. I think this is strong over-interpretation. Each of the clusters also contains nucleation and non-nucleation days. Thus, small humps may be induced by either one and cannot be used to derive criteria for NPF.

Line 398ff: Peak concentrations of nucleation mode are reached before 15:00h. Thus this increase in Aitken mode after 15:00h cannot inhibit NPF. More important than the particle concentration is the CS and the availability of condensing gases. The authors seem to assume that the latter are always similar, which is hardly justified.

Line 422: On 12th December is the wind speed low during NPF. It is high before. The authors claim that NPF is suppressed by local emissions on 19th November. Later on, they pretend that NPF occurs up to hundreds of kilometers upwind. Thus, the nucleation mode particles are transported to the site and cannot be influenced by local emissions.

Line 426: I wonder what else besides nitrate and sulfate can bind that much ammonia. This is hard to believe.

---

## Referee Comment (RC2) · Anonymous Referee #2 · 7 Aug 2018

In their work, Singla et al., have studied new particle formation events that have been observed at "high altitude" in India. During the measurement time they observed 47 NPF events. The data have been recorded using a ToF-ACSM, a WRAS and a CCNC, actually not the best set of instruments to study those type of processes. The authors calculated formation rate and growth rates for the most important events (10 events) and found quite high value. Despite that the conclusions of the paper and the scientific message is not very clear neither new.

After a careful read of the manuscript I would not recommend this work to be published in ACP at this current form. I would also add that there is no new information on what we

know already about NPF. Especially considering the fact that most of the conclusions don't have evidence in the manuscript.

Below I reported few major concerns.

In the title, the authors mention new particle formation at high altitude. After that, there is basically no reference anymore on the height of the sampling site. What does it mean high altitude? Are the measurements done in the free troposphere (FT) or in the planetary boundary layer (PBL)? This needs to be explained in the manuscript. It is fundamental to understand where are all these particles going after been formed. The lifetime in the free troposphere is larger than in the PBL making their climate impact very different. Without this information is actually useless to know the altitude especially because around the globe high and low altitude can mean totally different scenario. See for example the study conducted from Martine Collaud Coen et el., (2017) (https://www.atmos-chem-phys-discuss.net/acp-2017-692/). I would recommend the authors to look deeply in the newest literature focusing on new particle formation / nucleation at high altitude / free troposphere. Here just few papers (Rose et al., ACP 2017, Garcia et al., ACP 2015, Tröstl et al., JGR 2016, Venzac et al., PNAS, 2008)

In the abstract, but in general in the whole study, the authors' just report a series of numbers and characteristics of the most prominent NPF events. However there is no a clear message on the mechanism behind the NPF events or newer information on this process. Additionally, in the manuscript, there are basically no indications that support their conclusion. For example in the abstract they reported: ". . ..New particle formation (NPF) events were observed on 47 days and mainly associated with these north-easterly air masses and high SO2 emissions and biomass burning activities, while weaker or non-NPF days were associated with westerly air masses and relatively higher influence of local air pollution. . .." This is not confirmed by any of the data shown here.

Additionally, in the abstract, they reported:".. A closer examination of strong NPF

events showed that low relative humidity and solar radiation favored new particle formation". It is very common to see NPF correlating with radiation therefore nothing new but here as well there are no evidence that the Relative humidity is affecting NPF. Changed in RH could just reflect a change in air mass.

As mentioned already, the authors have a good set of instruments but unfortunately it is not the proper one to study NPF. They would need a chemical ionization mass spectrometer (CIMS) in order to measure sulfuric acid concentration or organic precursor concentration. Additionally, Since they based the conclusion on the SO2, would be important to have a monitor to measure this gas as well. Then it is extremely important to measure also physical propertied of the aerosol smaller than 5 nm, the most critical size when study NPF. Obviously it is difficult to measured all this parameters however would be good if the author would mention that in the text. For this kind of studies I believe that most of the conclusions can't be driven with the instruments used in this study. ToF-ACSM is a very good instrument but the cut off of the instrument is far too big to say something about nucleation. Here I would recommend the author to look for the latest study on NPF that have been conducted at the CLOUD chamber in CERN and their instrumentation.

Additional major comments:

Formation rates: when comparing no NPF day with NPF day would be important to calculate J5 for the non NPF days as well.

Growth rates: In order to explain a GR of 2.5 nm/h or more at such size, you would need a lots of H2SO4. This is probably not possible if you are not in a plume of an air mass with very high H2SO4 concentration. More considerations need to be done here as well and probably organics would play a major role as well.

Lines 268 to 275: the statements reported in these lines are not confirmed by the data. There is no evidence that the neutralized nature of aerosol favored nucleation. Additionally, the study mentions that NH3 plays a major role in those events. They

based this conclusion on the fact that during the non-events, ammonia is taken up by the acidic aerosol. Although the ammonia scavenging by the acidic aerosol might be real, there are no evidence the NH3 is driving NPF.

Additionally because of the nature of the ACSM the paragraph is 4.2 is not very useful in order to understand NPF. Also the diameter has to be defined. Is it Electrical mobility or aerodynamic diameter? If Electrical the cut off is around 40-50 nm otherwise around ∼75 nm. Additionally, if the aerodynamic diameter is converted the authors need to mention the density approximation that they used to make such conversion. Summarizing the ACSM can't say that the aerosol measured comes from NPF or not. As far as it is shown in this data, it can come from any sources. Therefore once more there are no evidences of the statements at the end of the paragraph (from line 286 – 292).

4.4 Cluster analysis The paragraph is quite long, vague and here as well the statements are not supported by the data. It is very difficult to believe that NPF is triggered by biomass burning without further evidence, chemical information and source appointments.

Finally, as the reviewer 1 mentioned already, most of the conclusions reported here are not supported by the data.

Minor Issue:

Paragraph 2.1 Measurement site. Here would be very helpful to explain what are the major sources around the site and if the site is situated in the planetary boundary layer, free troposphere or some layer in between. I also assume that this depend on the season and on the time of the day.

Line 230: Usually cold temperatures favour nucleation not hot. (Kikrby et al., Nature, 2011)

Line 233: How can be that a particle that is 5 nm big and grow at 2.5 nm/h reach 40

nm in 6 hours?

Line 255 :" The CS on NPF days was much lower than on non-NPF days (∼4.2 - 4.4*10-2 s-1). " The difference in CS between event and no event doesn't seems to be much lower. They are actually comparable. I would consider using more non-event days in the data analysis. One option would be to take an average of many them.
* * *

---

## Author Comment (AC1) · 9 Oct 2018

The paper reports new particle formation (NPF) events observed at the High Altitude Cloud Physics Laboratory (HACPL) in Mahabaleshwar in Western India. They measured nucleation and growth rates as well as condensation and coagulation losses. The authors further analyze under what conditions NPF occurs by using a detailed case analysis and a cluster analysis. The paper concludes that NPF is favored by low relative humidity and when air masses reach the site from central India. Using the NanoMap method it is estimated that NPF events occur several hundred kilometers upwind of the site.

The paper presents an interesting data set on NPF. However, many claims and conclusions I could not see and follow, e.g. the determination of CCN to NPF (details see below). Furthermore, many conclusions are not well established and over-interpreted. The paper needs substantial improvements following the detailed comments below.

Line 218: the sharp decrease occurs between 4-6h according to Figure 3a and there-fore before sunrise (Figure 2). This looks like a change of air mass as also seen in the change of hygroscopicity. It would also be more helpful to present absolute humidity in Figure 2 to check for air mass changes.

Response:

Thank you for the detailed check of the timing of events and suggestion to present the absolute humidity. We agree that there is some difference in the timing of events. Indeed, the decrease in total Aitken and accumulation mode number concentration that takes place until 08:00 in the morning is not caused by an increase in boundary layer height, since sunrise is not until 07:30, which is too late.

Absolute humidity (see figure below for 12[th] December, 2016) unfortunately could not add any any additional information in to the analysis. Nevertheless, the decrease in concentration between 05:00 and 08:00 is not due to a change of air mass, since the air mass trajectories (extracted using HYSPLIT model) and wind directions (figure given below) are stable throughout the day. Rather it is an effect of decreasing pollution within the same air mass. We have changed the text accordingly in the revised manuscript to make this more clear.

[Figure]

Figure: Diurnal variation of absolute humidity on 12th, 14th and 19th December, 2016.

Figure: Diurnal variation of absolute humidity on 12th, 14th and 19th December, 2016.

[Figure]

Figure: HYSPLIT airmass back trajectories (Hourly; left panel) and diurnal variation of meteorological parameters (right panel) on 12th December, 2016. **Date and time given in back trajectory plot is in UTC.**

To check whether absolute humidity is a sole marker for air mass change, we have checked the diurnal variation of absolute humidity on 14th and 19th December as well. It was found to vary along the day on 14th December but the airmass back trajectory and wind direction appears to be constant (figure given below). However on 19th December, the absolute humidity varied during daytime consistent with the change in airmass and wind direction (figure given below). The absolute humidity showed variation

same airmass (12th and 14th December, 2016) as well as different airmass. This is why absolute humidity may not be considered as a primary indicator for airmass change as it gets affected by evaporation, transpiration, condensation, precipitation, moisture, etc. within a same air mass.

[Figure]

Figure: HYSPLIT airmass back trajectories (Hourly; left panel) and diurnal variation of meteorological parameters (right panel) on 14th December, 2016. **Date and time given in back trajectory plot is in UTC.**

[Figure]

**Figure:** HYSPLIT airmass back trajectories (Hourly; left panel) and diurnal variation of meteorological parameters (right panel) on 19[th] December, 2016. **Date and time given in back trajectory plot is in UTC.**

Line 225: sunrise is at 7.30h and NPF is seen at 11h. This is not 60-90 minutes later. Probably you mean estimated start of NPF.

Response: The reviewer is right that we meant estimated start of NPF at 1.5 nm diameter. We only have measurements starting from 5.14 nm diameter, i. e. almost 2 hours after the formation of 1.5 nm diameter particles.

We have clarified in the manuscript that the formation of 1.5 nm diameter particles takes place around 09:00 to 09:30, almost 2 hours after sunrise at 07:30. And that the NPF is not visible in our size distribution measurements until around 11:00 since it takes approximately 2 hours for the particles to grow from 1.5 nm diameter to the lowest detectable size of 5.14 nm diameter in our size distribution measurements.

Line 229ff: What time period do you take for the average values? By the way I do not consider 20-24 C as hot.

Response: The solar radiation value given earlier was for the nucleation time (11:00-12:00 hrs). The values of wind speed, relative humidity were given as the average value during 12:00 to 17:00 hrs. To match the time periods, the parameters are now given as the average between 12:00 and 17:00 hrs.

We accept the temperature between 20-24 °C is not considered as hot. Therefore, we will remove the term 'hot' and revise the statement as "The less humid and stable weather condition tends to favor the enhancement of atmospheric nucleation (Hamed et al., 2011)."

Line 233: With a GR of 2.5 nm/hr it is impossible to reach 40 nm in 6 hours.

Response: The authors accept that it is impossible to reach 40 nm in 6 hours. Since we have calculated the growth rate in the size range 5-25 nm, we cannot derive the above conclusion. Now we have calculated the growth rate in the size range 5-900nm and accordingly revised the statement as "The particles were continuously formed at 5 nm diameter for a minimum of 2 hours and then grew at a rate of 5.1 nm hr$^{-1}$."

Line 248: How do you know that sulfuric acid is driving nucleation? It could also be organics. Furthermore, it could also be other parameters like availability of nucleating species.

Response: The authors accept that there could be other parameters also. It may not be correct to argue the role of sulfuric acid as there are no observations of sulfur dioxide or sulfuric acid during the measurement period. With this line we tried to depict one of the possible way that how relative humidity can affect the NPF formation. Hamed et al., 2010 have showed that high relative humidity tends to cut down the availability of atmospheric oxidants (OH). Based on this study, we have also rephrased this line as "High relative humidity results in lowering of OH radical (important oxidant) in the ambient air which in turn limits the availability of nucleating species (sulfuric acid, organic vapors etc.)"

Line 195: this GR is for particles larger than 5 nm. However, for smaller particles GR is usually quite slower due to the Kelvin effect (see Kulmala 2013).

Response: The reviewer is right that GR is normally lower for sub-5-nm diameter particles than for particles >5 nm diameter. However, in the absence of sub-5-nm diameter particles, we have approximated the GR from 1.5 nm diameter to 5 nm diameter with the average GR found for particles

between 5 and 25 nm diameter. Hence, this might be a systematic overestimation of GR and therefore an underestimation of time needed for growth between 1.5 and 5 nm diameter. Besides, for individual days the time needed for growth between 1.5 and 5 nm diameter might be very different from the average time (1.5 h), hence again leads to bias.

On most of the time only a small error in comparison to other uncertainties in parameters calculated in this study, since the time needed for growth between 1.5 and 5 nm diameters is only 1.5 h. Nevertheless, we have stressed in the manuscript that the 1.5 h is only an approximation to the realistic value, since we lack data below 5 nm diameter.

Line 256: The CS is only a factor 2 higher on non-NPF days. This is not a much higher value.

Response: The authors agree that CS value on non-NPF day is not a much higher value. Further as suggested by Reviewer 2, we have also calculated the average CS value for all non-NPF days as $4.2\pm1.9$ $*10^{-2}$ s$^{-1}$.

Based on calculated average value of the CS for strong (now referred to as Type 1a NPF days) NPF and non-NPF days respectively within one standard deviation, we have come to the conclusion that the CS on non-NPF days is slightly higher, and that this is one of the contributing, but may not be a crucial reason for the appearance of strong NPF days.

Line 263-264: 9:30 to 11:00h is the assumed time of formation of particles, while 11:00 to 13:30 h is the observed formation. Make this clear.

Response: Thank you for the query. 9:30 to 11:00 h is the possible time for nucleation (formation of 1.5nm particle) and 11:00 to 13:30 is the observed formation because of the lack of data below 5.14 nm. Therefore, we observed the nucleation after 11:00 h. We have now clarified this statement in the manuscript.

Line 266: The decrease in aerosol mass happens before. When NPF starts, PNC and aerosol mass are already low.

Response: In our study, we have observed that conditions with low PNC and low aerosol mass concentration favored NPF formation. We accept that PNC and aerosol mass was already low and therefore rephrased the line as "During the first time window, aerosol mass concentration and PNC were found to be at its minimum."

Line 269: What do you mean by "formation of new particles is governed by the nature of pre-existing aerosol particles"?

Response: Thanks for the comment. Authors would like to rephrase the above mentioned sentence as "formation of new particles may be governed by the chemical nature of pre-existing aerosol particles". A study by Pikridas et al., 2014 has reported that nucleation events occur only when particles are neutral in nature as compared to NH$_3$ limited nucleation in sulfate-rich environments. Following the same concept, we have also calculated the particle acidity for NPF and non-NPF days. The calculated particle acidity shows acidic nature of particles on non-NPF day while neutral nature on NPF day. Based on this we have concluded that the particle acidity may also be one of the influencing factors for NPF formation.

Line 270: Is this the equivalent ammonium to anion ratio? What time interval does this cover? Only after 9h?

Response: Yes, this is the equivalent ammonium to anion ratio. This is the average value for the time period 09:00 to 11:00 hrs. We have added this information in the revised manuscript.

Line 278: organics increase before inorganics. How do you interpret that?

Response: In line 278, Authors intend to say that the mass concentration of aerosol increased 2 hours later than the increase in Aitken mode volume concentration (13:30 to 14:30 hrs.). The high organics concentration after 14:00 hrs is possibly due to primary emissions as secondary species like sulfate does not show any increase within the same time window. This information will be added in the revised manuscript.

Line 286: Much of the increase in aerosol mass is by HOA and BBOA and not due to additional mass from NPF. In Figure 1 it is seen that the concentration of particles 50-200nm increases after 14h. What can now be attributed to this increase and to particles grown in from NPF? What is the cut-off of the ACSM?

Response: The authors accept that the increase in aerosol mass may be contributed by HOA and BBOA. The increase in the concentration of particles 50-200 nm around 15:00 hrs may be attributed to the transition of sampling site from boundary layer influence to free troposphere (see graph below). The standard deviation of horizontal wind direction above 12.5 indicates the influence of atmospheric boundary layer. The dilution in the 50-200 nm particles (figure below) was observed when the site got influenced by boundary layer (13:00 to 15:00 h). The further increase in 50-200 nm particles after 16:00 hrs is possibly due to primary emissions (enhancement due to traffic emission and biomass burning), which is also reflected by the increase in concentration of organics (as mentioned in comment above). The further increase in 50-200 nm size particles after 18:00hrs is contributed by NPF.

[Figure]

Line 289: this is speculation. CCN starts to increase already at 12h, one hour after observing NPF at 5 nm. How can these small particles be activated so early? Could it also be that particles in the 50-200 nm band start to grow due to condensation of organics and become CCN active?

Response: We agree with the reviewer that the non-NPF particles 50-200 nm diameter have contributed to the short-time peak of CCN around 12:00, and also to the more pronounced increase between 14:00 and 18:00. After this time period, and later during the night hours, the relative increase in CCN is due to an increase of NPF grown particles, since a large part of the NPF population has grown to sizes well above 40 nm diameter. We have changed the text in the manuscript accordingly in line with the comment of the reviewer.

Section 4.3: There are several issues with the estimation of NPF to CCN. The authors say that they compare time window 2 with window 1 and that GMD > 40 nm. In Figure 1 these criteria cannot be met but calculations are given. The notation given for the different parameters (CCN/POA, etc.) is unclear

and not consistent. I had to consult the original publication to understand it. All values for delta (CCN/CN) are positive. I have a hard time to believe that the fraction of CCN is always larger shortly after nucleation (11:00-13:30h) than in the aged air mass when no nucleation mode CN are measured yet. This would mean that an extremely large fraction of nucleation mode particles (most particles are still <40 nm) activates. I also doubt that CCN/POA is a good measure for primary contributions to CCN. Along a trajectory POA decreases while CN may not change much. This pretends an increase in CCN fraction that is not real. As already mentioned above an increase in CCN may also be due to growth of 50-200nm particles. How do the authors account for this?

Response: Here we are somewhat confused with the comment about time window 2. Time window 2 is not around 11:00-13:30 hrs. From figure 1 it can be clearly seen that time window 2 is around midnight. The chosen criteria for evaluating the effect of NPF on CCN are followed for the two windows of Figure 1. The dashed line in Figure 1 represents the particle diameter of 40 nm and black dots represent the calculated GMD of particles. At this time, GMD is much higher than 40 nm diameter. But, since there is a chance to miss time window 2 in figure 1, we have also stressed the timing of time window 2 in the manuscript text.

[Figure]

We are limited with the direct measurement of primary emission tracers. Though we had EC measurements but the EC data was not available for all the NPF days. This is why POA has been chosen as a proxy for primary emission. We have compared the EC data with POA and it showed fair correlation (figure given below). The assumption of POA as primary tracer may add some uncertainty in the calculation but this error is not expected to be huge.

[Figure]

The parameter CCN/POA was calculated to assess the effect of fresh anthropogenic sources on the activation ratio (CCN/CN). Biomass burning and vehicular emissions are the major anthropogenic sources at this site (Mukherjee et al., 2018). Therefore POA was chosen as a marker of fresh emissions. The details of the methodology have been added in the revised manuscript. Further, the authors accept that all values for delta (CCN/CN) cannot be positive. Actually all the delta (CCN/CN) values were calculated as negative in this study and the negative sign was somehow missed in the header of the column in Table 2.

Section 4.4: this section contains many claims that can hardly be seen in the Figures and small variations in measurements are overly interpreted. Line 368: I do not see the decrease in Aitken mode concentration during increase of nucleation mode particles.

Response: Indeed, the increase mentioned for NPF particles between 11:00 and 14:00 in manuscript is not coinciding simultaneously with a decrease of Aitken mode particles. The decrease of Aitken mode particles is actually taking place earlier, already starting from around 09:00. We have not been clear about this, but have now explained it in the manuscript.

The authors would also like to mention that the Aitken mode number concentration decreased by 20% (from 2752 cm$^{-3}$ to 2194 cm$^{-3}$) during the increase of nucleation mode particles.

Line 371: Onset of nucleation is usually around 9:00h. I do not see a sudden increase in Aitken mode particles. In addition, average Aitken mode concentration at that time is similar in cluster C2 and C3.

Response: Thank you for the comment. We have tried to answer the mentioned comment and next comment together as they are linked.

Line 372: It says: The diurnal variation of Nnuc particles exhibited high concentration during morning, afternoon and evening hours". I do not see high concentrations throughout the day. I also do not see the simultaneous peaks of nucleation and Aitken modes. There are time shifts.

Response: Both this comment and the previous one will be answered here. We are again sorry for over interpretation and over simplification of the timing of events.

It is enough that we rephrase the manuscript to say:

1. The PNC is in general more variable throughout the day for clusters C4 and C5 and nucleation and Aitken mode particles, which is indicating the influence of local pollution sources.

2. There is a clear morning rush hour traffic peak in the morning hours for C4 and C5 (figure S2), stronger than for C1, C2, and C3 likely contributing to inhibit the onset of formation, which normally takes place around 09:00 at the measurement site.

3. However, this does not rule out the onset of formation after the morning hours, as the Aitken mode concentration is decreasing in C4 and C5. We have no claim for the following statement, but the transport of air from foreign areas in clusters C4 and C5 might not allow for production of low volatile vapors needed to initiate NPF after the morning hours.

Line 398ff: Peak concentrations of nucleation mode are reached before 15:00h. Thus this increase in Aitken mode after 15:00h cannot inhibit NPF. More important than the particle concentration is the CS and the availability of condensing gases. The authors seem to assume that the latter are always similar, which is hardly justified.

Response: Thanks for the comment. Based on the influence of boundary layer at sampling site, we accept that the increase in Aitken mode after NPF cannot inhibit NPF. We have now removed this concept from section 4.4.

Line 422: On 12th December is the wind speed low during NPF. It is high before. The authors claim that NPF is suppressed by local emissions on 19th November. Later on, they pretend that NPF occurs up to hundreds of kilometers upwind. Thus, the nucleation mode particles are transported to the site and cannot be influenced by local emissions.

Response: Thanks for the comment. To assess the effect of local emissions on the nucleation process, we have also studied the influence of BL/FT at the sampling site. The formation of 1.5 nm particles is expected to occur at 9:00-10:00 hrs when the site is influenced by FT. The figure provided below shows that the site comes under the influence of BL after 10:00 hrs on 19[th] November, 2016. We believe that the effects of local emissions are pronounced only when the site is influenced by boundary layer. This phenomenon affected the intensity of the NPF on that day. We have also checked for the influence of BL/FT for other strong (now referred as Type 1a following Dal Maso et al., 2005), weak (now referred as Type 1b following Dal Maso et al., 2005) and non-NPF days and as a representative, 2 days of each type is given below.

[Figure]

Figure: Diurnal variation of standard deviation of horizontal wind direction on 19th November, 2016.

[Figure]

Figure: Diurnal variation of standard deviation of horizontal wind direction for 2 representative type 1a (upper panel), type 1b (middle panel) and non-NPF days.

The figure above shows that influence of FT at the sampling site favours nucleation and vice-versa.

Line 426: I wonder what else besides nitrate and sulfate can bind that much ammonia. This is hard to believe.

Response: The high molar ratio of ammonium to inorganic ions ~1.4 was observed on 19th November 2016 at the time of nucleation. During October-November, the plantation of strawberry is carried out in this region for which the fertilizer (mainly urea) is dissolved in water and is sprayed (high in ammonium) more often. This could be one of the possible reasons why we are getting high ammonium during the month of November.

---

## Author Comment (AC2) · 17 Oct 2018

In their work, Singla et al., have studied new particle formation events that have been observed at "high altitude" in India. During the measurement time they observed 47 NPF events. The data have been recorded using a ToF-ACSM, a WRAS and a CCNC, actually not the best set of instruments to study those type of processes. The authors calculated formation rate and growth rates for the most important events (10 events) and found quite high value. Despite that the conclusions of the paper and the scientific message is not very clear neither new.

After a careful read of the manuscript I would not recommend this work to be published in ACP at this current form. I would also add that there is no new information on what we know already about NPF. Especially considering the fact that most of the conclusions don't have evidence in the manuscript.

Response: Authors would like to thank the reviewer for his comments and suggestions. We admit that the previous version of the manuscript is not very clear enough with conclusions, and we have taken steps to make the conclusions clearer and removed non-conclusive facts. We would like to state that we don't claim to search for the mechanisms behind NPF formation, but wish to convey that that there are environmental factors favorable for the onset of formation and continued growth of NPF particles. Without this information it is difficult to undertake a study looking for the mechanism behind NPF formation in this region. Hence, our study is an important preparation for future studies, where we can select proper instrumentation and investigate the mechanisms leading to NPF and growth in this region. We also understand that GR and FR analysis has been performed before in other regions and that this is not novel. But, not many reports are there over this geographical area and we have no idea previously where, when and how NPF are formed in this region. Since how NPF are formed are not known, and models do not predict these NPF satisfactory in new regions where measurements have not been performed before, more information from new sites are necessary. And we made some new interesting analyses which are novel, for example the NanoMap analysis and cluster analysis highlighting the importance of aged continental airmass.

Below I reported few major concerns

In the title, the authors mention new particle formation at high altitude. After that, there is basically no reference anymore on the height of the sampling site. What does it mean high altitude? Are the measurements done in the free troposphere (FT) or in the planetary boundary layer (PBL)? This needs to be explained in the manuscript. It is fundamental to understand where all these particles going after are been formed. The lifetime in the free troposphere is larger than in the PBL making their climate impact very different. Without this information is actually useless to know the altitude especially because around the globe high and low altitude can mean totally different scenario. See for example the study conducted from Martine Collaud Coen et el., (2017) (https://www.atmos-chem-phys-discuss.net/acp-2017-692/). I would recommend the authors to look deeply in the newest literature focusing on new particle formation /

nu-cleation at high altitude / free troposphere. Here just few papers (Rose et al., ACP 2017, Garcia et al., ACP 2015, Tröstl et al., JGR 2016, Venzac et al., PNAS, 2008)

Response: Authors thank the reviewer for suggesting related references. As suggested, an attempt was made to see whether our observation site is influenced by planetary boundary layer or free troposphere. Following the methodology of Rose et al., 2017 (as suggested by the Reviewer), we have calculated the standard deviation of horizontal wind direction. It was observed that during strong (now referred to as type 1a) NPF events, the site stays in free troposphere at the time of NPF formation and influenced by boundary layer only after NPF (~ 13:00 hrs). In case of weak NPF days, the observation site comes under the influence of boundary layer at the time of nucleation which is probably affecting the intensity of NPF and subsequent growth of newly formed particles. During non-NPF days, the observation site comes under the influence of atmospheric boundary layer generally during the morning hour (~ 10:00 hrs) which is expected to inhibit the nucleation process. The diurnal variation of standard deviation of horizontal wind direction for few days is given below for reference.

[Figure]

Bianchi et al., 2016 have given the observational evidence that highly oxygenated molecules (HOMS) can initiate the NPF process in addition to sulfuric acid–ammonia nucleation at high altitude sites. We have also observed the high degree of oxygenation for LV-OOA (low volatile oxygenated organic aerosol) during this time period (f44 ~ 0.4 Mukherjee et al., 2018). The diurnal variation of horizontal wind direction (graph below) during strong NPF days show that the site is in free troposphere up to 12:00 hrs. Since the site is in free troposphere, we would expect minimum influence of local activities. Therefore the formation of strong NPF at our site under the influence of free troposphere indicates the likely presence of highly oxygenated molecules (HOMS) at our site which may be acting as a fuel for NPF. However, without strong observational evidence, this argument is very difficult to establish with current measurement limitation.

In the abstract, but in general in the whole study, the authors' just report a series of numbers and characteristics of the most prominent NPF events. However there is no a clear message on the mechanism behind the NPF events or newer information on this process. Additionally, in the manuscript, there are basically no indications that support their conclusion. For example in the abstract they reported: ": : :.New particle formation (NPF) events were observed on 47 days and mainly associated with these north-easterly air masses and high SO2 emissions and biomass burning activities, while weaker or non-NPF days were associated with westerly air masses and relatively higher influence of local air pollution: : :." This is not confirmed by any of the data shown here.

Response: It is true that we don't make any claim to present a mechanism for the new particle formation. However, we have focused on to present the environmental conditions favorable for NPF, in this case $SO_2$ emissions and biomass burning activities in the region. We have proven this circumstantial evidence of NPF dependent on this regional pollution in chapter 4.4. However, we will summarize our findings from chapter 4.4 in the chapter (it is relatively long) to make it clearer about our conclusions from this chapter.

Additionally, in the abstract, they reported:". A closer examination of strong NPF events showed that low relative humidity and solar radiation favored new particle formation". It is very common to see NPF correlating with radiation therefore nothing new but here as well there is no evidence that the Relative humidity is affecting NPF. Changed in RH could just reflect a change in air mass.

Response: Thanks for the comment. Although it is well known that solar radiation often favors new particle formation at other sites globally, it is worth mentioning also for this specific geographic region. We have already shown the variation of RH for NPF and non-NPF day in Figure 2. High RH on non-NPF day itself is the evidence that RH is affecting NPF formation. Hamed et al., 2011 showed that high relative humidity tends to limit the concentration of available precursor gases by reducing the oxidant concentration (OH). Moreover, change in RH or absolute humidity may reflect a change in air mass but the relationship between them is not linear. There are many factors like evaporation, transpiration, condensation, precipitation and moisture etc. which can change the RH or absolute humidity within a same air mass. To see the possible influence of RH on NPF, other factors like airmass and PBL were also looked into for 14th December 2016. It can be seen from the figure (below) that the hourly HYSPLIT backward trajectory and wind direction (Figure 2 of manuscript) hardly shows any change in airmass throughout the day and the site resides in free troposphere (evident from the diurnal variation of $\sigma_\theta$). Still we are unable to see any NPF activity. This depicts the effect of high humidity on NPF formation.

[Figure]

[Figure]

As mentioned already, the authors have a good set of instruments but unfortunately it is not the proper one to study NPF. They would need a chemical ionization mass spectrometer (CIMS) in order to measure sulfuric acid concentration or organic precursor concentration. Additionally, Since they based the conclusion on the SO2, would be important to have a monitor to measure this gas as well. Then it is extremely important to measure also physical propertied of the aerosol smaller than 5 nm, the most critical size when study NPF. Obviously it is difficult to measured all this parameters however would be good if the author would mention that in the text. For this kind of studies I believe that most of the conclusions can't be driven with the instruments used in this study. ToF-ACSM is a very good instrument but the cutoff of the instrument is far too big to say something about nucleation. Here I would recommend the author to look for the latest study on NPF that have been conducted at the CLOUD chamber in CERN and their instrumentation.

Response: Thanks for the suggestion. Again, we would like to point out that it is not our aim to provide mechanisms on the onset of NPF. Nevertheless, we agree that we should discuss the areas which we are lacking at present in the revised manuscript. Author also agrees that CIMS is one of the ideal instruments to study the NPF events but here we have tried to investigate the NPF formation with best possible resources available. We are using ACSM (cut off size ~40nm) as a proxy to see the change in mass concentration due to NPF formation, if any. The intention is not to draw any big conclusion using ACSM data rather to highlight the possible effect of NPF event on ambient aerosol mass loading.

Additional major comments:

Formation rates: when comparing no NPF day with NPF day would be important to calculate J5 for the non NPF days as well.

Response: Thanks for the comment. However, we cannot calculate J5 for the non-NPF days since there is no formation of new particles.

Growth rates: In order to explain a GR of 2.5 nm/h or more at such size, you would need lots of H2SO4. This is probably not possible if you are not in a plume of an air mass with very high H2SO4 concentration. More considerations need to be done here as well and probably organics would play a major role as well.

Response: We agree with the reviewer that the growth rate of 2.5nm/h at nucleation size mode has to be supported by both high $H_2SO_4$ and highly oxygenated organics as well. In Mukherjee et al., 2018 we have reported LV-OOA with f44 ~0.4 which is quite high. It is also proven from the cluster analysis and concentration weighted trajectory (CWT) analysis that the site is influenced by long range transport from Central India region. The chances of availability of highly oxygenated molecule are quite high which may serve as a fuel to NPF formation under suitable meteorological conditions. The CIMS observation will be ideal to check the role of organics in NPF.

Lines 268 to 275: the statements reported in these lines are not confirmed by the data. There is no evidence that the neutralized nature of aerosol favored nucleation. Additionally, the study mentions that NH3 plays a major role in those events. They based this conclusion on the fact that during the non-events,

ammonia is taken up by the acidic aerosol. Although the ammonia scavenging by the acidic aerosol might be real, there are no evidence the NH3 is driving NPF.

Response: The nature of pre-existing particles was calculated in terms of particle acidity. The authors have provided the value of particle acidity for both NPF and non-NPF days as 1.08 and 0.88-0.90 respectively. The study by Pikridas et al., 2014 has reported that nucleation events occurred only when particles were neutral while lack of $NH_3$ can limit nucleation in sulfate-rich environments. We are not deriving the conclusion that the NPF is driven primarily by the ammonia, but the availability of ammonia can be one of the factors which may drive NPF formation.

Pikridas, M., Riipinen, I., Hildebrandt, L., Kostenidou, E., Manninen, H., Mihalopoulos, N., ... & Pandis, S. N. (2012). New particle formation at a remote site in the eastern Mediterranean. *Journal of Geophysical Research: Atmospheres*, *117*(D12).

Additionally because of the nature of the ACSM the paragraph is 4.2 is not very useful in order to understand NPF. Also the diameter has to be defined. Is it Electrical mobility or aerodynamic diameter? If Electrical the cut off is around 40-50 nm otherwise around 75 nm. Additionally, if the aerodynamic diameter is converted the authors need to mention the density approximation that they used to make such conversion. Summarizing the ACSM can't say that the aerosol measured comes from NPF or not. As far as it is shown in this data, it can come from any sources. Therefore once more there are no evidences of the statements at the end of the paragraph (from line 286 – 292).

Response: The quoted particle diameter range measured by ACSM (~40-1000 nm for $PM_1$ lens) is vacuum aerodynamic diameter. Measurements of the particle transmission were made with material of known density. For our instrument we have used $NH_4NO_3$ and the density of 1.72 g cm$^{-3}$ was used in the conversion. In the case of $NH_4NO_3$ (non-spherical and crystalline), an additional correction factor was used to convert the material density to the particle density which is called as shape factor. The shape factor for our instrument was determined by comparing particle time-of-flight in an AMS system for the material with an unknown shape factor with a solid spherical particle (i.e. polystyrene spheres).

It is true that the chemical compounds measured by the ACSM come from a variety of sources. But, as we have suggested also to reviewer no. 1: the mass fraction originating from NPF grown particle is increasing with increasing time after onset of formation, and during the night, a considerable part of ACSM measured chemical mass comes from NPF grown particles compared to other particle sources. This can be evidenced by observing the grown NPF particle mode during evening and night hours. We have clarified this in the manuscript.

4.4 Cluster analysis The paragraph is quite long, vague and here as well the statements are not supported by the data. It is very difficult to believe that NPF is triggered by biomass burning without further evidence, chemical information and source appointments.

Finally, as the reviewer 1 mentioned already, most of the conclusions reported here are not supported by the data.

Response: Thanks again for reminding us about making the goals clear for our data. We would like to repeat that we don't claim to search for the mechanisms behind NPF formation, but wish to convey that that there are environmental factors favorable for the onset of formation and continued growth of NPF

particles. Without this information it is difficult to undertake a study looking for the mechanism behind NPF formation in this region. Hence, our study is an important preparation for future studies, where we can select proper instrumentation and investigate the mechanisms leading to NPF and growth in this region. We have also tried to explain the cluster analysis on the basis of aerosol chemical composition, particularly in terms of organic aerosol components (HOA, BBOA and OOA).

Minor Issue:

Paragraph 2.1 Measurement site. Here would be very helpful to explain what are the major sources around the site and if the site is situated in the planetary boundary layer, free troposphere or some layer in between. I also assume that this depend on the season and on the time of the day.

Response: We have already described the major sources around our site in Mukherjee et al., 2018. We have added the major sources around the observation site in the section 2.1 of the revised manuscript. We have added information on whether the site is situated in the planetary boundary layer or free troposphere. We have given few graphs related to this comment in earlier response.

Line 230: Usually cold temperatures favour nucleation not hot. (Kikrby et al., Nature, 2011)

Response: Author agrees with the comment. The line is rephrased and may read as "This less humid and stable weather condition tends to favour the enhancement of atmospheric nucleation (Hamed et al., 2011)."

Line 233: How can be that a particle that is 5 nm big and grow at 2.5 nm/h reach 40 nm in 6 hours?

Response: The authors accept that it is impossible to reach 40 nm in 6 hours. Since we were calculating the growth rate in the size range 5-25 nm, we cannot derive the above conclusion. Now we have calculated the growth rate in the size range 5-900 nm and accordingly revised the statement as "The particles were continuously formed at 5 nm diameter for a minimum of 2 hours and then grew at a rate of 5.1 nm $hr^{-1}$."

Line 255 :" The CS on NPF days was much lower than on non-NPF days ( 4.2 - 4.4*10-2 s-1). " The difference in CS between event and no event doesn't seems to be much lower. They are actually comparable. I would consider using more non-event days in the data analysis. One option would be to take an average of many them.

Response: The authors agree that CS value on non-NPF day is not a much higher value. As suggested, we have also calculated the average CS value for all non-NPF days as $4.2 \pm 1.8*10^{-2}$ $s^{-1}$. Indeed, a factor 2 is not much higher. Based on calculated average value of the CS for strong NPF and non-NPF days within one standard deviation, we have come to the conclusion that CS on non-NPF days is slightly higher, and that this is contributing, but maybe not a crucial reason for the appearance of strong NPF days.